# Comprehensive Identification and Functional Analysis of Stress-Associated Protein (SAP) Genes in Osmotic Stress in Maize

**DOI:** 10.3390/ijms232214010

**Published:** 2022-11-13

**Authors:** Qiankun Fu, Huaming Duan, Yang Cao, Yan Li, XiaoLong Lin, Haowan Pang, Qingqing Yang, Wanchen Li, Fengling Fu, Yuanyuan Zhang, Haoqiang Yu

**Affiliations:** 1Key Laboratory of Biology and Genetic Improvement of Maize in Southwest Region, Ministry of Agriculture, Maize Research Institute, Sichuan Agricultural University, Chengdu 611130, China; 2College of Life Science & Biotechnology, Mianyang Teachers’ College, Mianyang 621000, China

**Keywords:** A20/AN1 zinc finger, stress associated proteins, expression pattern, maize

## Abstract

Stress-associated proteins (SAPs) are a kind of zinc finger protein with an A20/AN1 domain and contribute to plants’ adaption to various abiotic and biological stimuli. However, little is known about the *SAP* genes in maize (*Zea mays* L.). In the present study, the *SAP* genes were identified from the maize genome. Subsequently, the protein properties, gene structure and duplication, chromosomal location, and *cis*-acting elements were analyzed by bioinformatic methods. Finally, their expression profiles under osmotic stresses, including drought and salinity, as well as ABA, and overexpression in *Saccharomyces cerevisiae* W303a cells, were performed to uncover the potential function. The results showed that a total of 10 *SAP* genes were identified and named *ZmSAP1* to *ZmSAP10* in maize, which was unevenly distributed on six of the ten maize chromosomes. The ZmSAP1, ZmSAP4, ZmSAP5, ZmSAP6, ZmSAP7, ZmSAP8 and ZmSAP10 had an A20 domain at N terminus and AN1 domain at C terminus, respectively. Only ZmSAP2 possessed a single AN1 domain at the N terminus. ZmSAP3 and ZmSAP9 both contained two AN1 domains without an A20 domain. Most *ZmSAP* genes lost introns and had abundant stress- and hormone-responsive *cis*-elements in their promoter region. The results of quantitative real-time PCR showed that all *ZmSAP* genes were regulated by drought and saline stresses, as well as ABA induction. Moreover, heterologous expression of *ZmSAP2* and *ZmSAP7* significantly improved the saline tolerance of yeast cells. The study provides insights into further underlying the function of ZmSAPs in regulating stress response in maize.

## 1. Introduction

Plants constantly encounter biotic and abiotic stresses from their surroundings. Consequently, plant growth, development and production are restricted by these stimuli, such as drought and salt stress [1,2]. To avoid adverse conditions, plants have evolved multifaceted strategies at morphology, physiology, and molecular levels to perceive, transfer and activate signal transduction to response stresses [3,4]. Among them, stress-related genes play pivotal roles in stress response. The stress-associated proteins (SAPs), a family of zinc-finger proteins with A20/AN1 domains, were first discovered in humans and *Xenopus laevis* and played key roles in innate immunity and cell death [5,6,7]. Most SAPs possess a typical SAP domain containing both A20 and AN1 domains presented in the N-terminal and C-terminal, respectively, and were separated by a variable stretch of amino acids [8]. Subsequently, the SAPs have been identified in all eukaryotes and confirmed as novel regulators in plant abiotic stress response [9,10,11].

In plants, OSISAP1 was first identified as A20/AN1 zinc-finger protein from rice and induced by multiple stresses, including cold, desiccation, salt, submergence and heavy metals, as well as injury [10]. Meanwhile, transgenic tobacco with an *OSISAP1* gene showed enhancements in the tolerance of cold, dehydration and salt [10]. Thereafter, using the *OsSAP1* sequence as a reference, 18 and 14 *SAP* genes (named *OsSAP* and *AtSAP*) were identified from the genome of rice and *Arabidopsis*, respectively, through a sequence similarity approach blast [11]. Most *OsSAPs* and *AtSAPs* have been confirmed to regulate abiotic stress responses such as drought, salt, and temperature [12,13,14,15,16,17,18]. Previous studies also showed that *SAPs* from wheat (*TaSAP5*), banana (*MusaSAP1*), *Populus trichocarpa* (*PtSAP13*), *Aeluropus littoralis* (*AlSAP*) and *Leymus chinensis* (*LcSAP*) positively regulate drought and salt tolerance [19,20,21,22,23].

In addition to abiotic stress response, plant SAPs are found to serve as an important hub to mediate disease resistance and development. For instance, Pha13 containing A20/AN1 zinc finger domains and its homologs AtSAP5, AtSAP9 and OsSAP1 are also involved in virus resistance, basal resistance against pathogen infection and compromising innate immunity [24,25,26]. The AtSAP9, PagSAP11 of poplar, and PpSAP1 of *Prunus persica* control flowering time, branching of lateral shoots, and cell expansion [25,27,28]. The OsDOG and ZFP185 are A20/AN1 zinc-finger proteins, negatively regulating cell elongation and size in rice [29,30]. Abscisic acid (ABA) is one of the most important phytohormones and plays key roles in plant growth, development and stress response [31,32]. Available reports showed that SAPs likewise mediate ABA signaling. Rice *OsiSAP7* negatively regulates ABA signaling to impart sensitivity to water-deficit stress [33]. However, ZFP185 modulates the expression of ABA biosynthesis-related genes and alters ABA content in plants to negatively regulate stress tolerance [30]. AtSAP9 is involved in the ABA-dependent regulation of downstream ABA-responsive genes and confers hypersensitivity to ABA of overexpressing plants [25].

Previous studies indicate that *SAPs* have emerged as promising candidates for improving stress tolerance and growth during unfavorable conditions in plants. As one of the most important crops, maize is a key factor in developing the national economy and maintaining food security [34]. However, the *SAP*s of maize remain poorly understood. In this study, the *ZmSAPs* were first identified in the maize genome and comprehensively characterized for protein properties, gene structure and duplication, chromosomal locations, *cis*-acting regulatory elements and tissue expression profiles. Additionally, the expression profiles of *ZmSAPs* under different abiotic stress and hormone induction were investigated by RT-qPCR. The function of *ZmSAPs* was validated in yeast. The study provides insights into the further underlying function of ZmSAPs and helps in understanding the known modes of SAPs action in plants.

## 2. Results

### 2.1. The ZmSAP Members in Maize

To identify ZmSAPs in maize, the amino acid sequences of 14 and 18 SAPs of *Arabidopsis* and rice were used as queries in local BLAST searches, respectively. Totally, 10 *SAP* genes were identified from the maize genome and named *ZmSAP1*~*ZmSAP10* (Table 1). The encoding sequences of *ZmSAPs* were 459 to 873 bp in length, encoding 152 to 290 amino acids (aa), with a molecular weight of 16.00 to 32.04 kDa, respectively. All ZmSAP proteins were predicted to be basic proteins with high theoretical isoelectric points ranging from 8.28 to 9.53. The grand average of hydropathicity (GRAVY) and instability indices of all ZmSAPs was <0 and >40, respectively, suggesting they were hydrophilic and instable proteins. Meanwhile, ZmSAPs showed no signal peptides and transmembrane region but were predicted to be nuclear localization.

### 2.2. Conserved Domains and Phylogenetic Analysis of ZmSAPs

The CDD analysis and sequence alignments showed that seven ZmSAPs, including ZmSAP1, ZmSAP4, ZmSAP5, ZmSAP6, ZmSAP7, ZmSAP8 and ZmSAP10, had an A20 domain at the N terminus and an AN1 domain at the C terminus, respectively. Only ZmSAP2 possessed a single AN1 domain at the N terminus. ZmSAP3 and ZmSAP9 both contained two AN1 domains without an A20 domain. Moreover, ZmSAP9 had two C2H2 domains (Figure 1A and Appendix A). Meanwhile, to explore the conserved motifs of ZmSAPs, these amino acid sequences were analyzed using the MEME tool. The results showed that ZmSAPs exhibited similar motif composition (Figure 1B). Among them, motif 1 and motif 3 were highly conserved and contributed to the A20 and AN1 domains of ZmSAP1, ZmSAP2, ZmSAP4, ZmSAP5, ZmSAP6, ZmSAP7, ZmSAP8 and ZmSAP10, respectively. In addition, there was a conserved motif 2 at the N terminus of these eight ZmSAPs behind the AN1 domain. Motif 4 and motif 5 were composed of two AN1 domains of ZmSAP3 and ZmSAP9, respectively.

To analyze the phylogenetic relationship between ZmSAPs and SAPs of *Arabidopsis* and rice, the amino acid sequences of 10 ZmSAPs, 14 AtSAPs and 18 OsSAPs were muti-aligned and used for phylogenetic tree construction. As shown in Figure 2, a total of 42 SAPs were divided into five branches. However, ZmSAPs were distributed in four branches besides group II (Figure 2 and Appendix A). The ZmSAP5 was branched in group I. The ZmSAP6, ZmSAP7 and ZmSAP10 were located in subgroup III. The ZmSAP1, ZmSAP2, ZmSAP4 and ZmSAP8 were located in subgroup IV. The ZmSAP3 and ZmSAP9 were branched in group V.

### 2.3. Chromosome Localization and Synteny Analysis

According to the information on the physical positions of ZmSAPs in maizeGDB, their chromosomal locations were visualized to ten maize chromosomes (Figure 3). *ZmSAPs* were located on six chromosomes. There were two *ZmSAP* genes on chromosome 1 (*ZmSAP1* and *ZmSAP2*), chromosome 2 (*ZmSAP3* and *ZmSAP4*), chromosome 4 (*ZmSAP5* and *ZmSAP6*) and chromosome 7 (*ZmSAP8* and *ZmSAP9*), respectively. The *ZmSAP7* and *ZmSAP10* were located on chromosome 5 and chromosome 9, respectively (Figure 3).

In addition, gene duplication analysis showed that *ZmSAP6* and *ZmSAP7* were identified as segmental replication, which belonged to the paralogous pair. Likewise, multi-species collinearity analysis spectra were constructed with *Arabidopsis* and rice. The results showed no orthologous pairs between maize and *Arabidopsis*, while fourteen orthologous pairs between maize and rice were identified as orthologs (Figure 3 and Appendix A), indicating that there was a high frequency of gene duplication between rice and maize in the process of evolution.

### 2.4. Gene Structure and Cis-Acting Elements of ZmSAPs

The exon–intron structure analysis showed that the genomic DNA (gDNA) sequence of *ZmSAP9* contained one intron. Other *ZmSAPs* had no intron and one exon. Among them, *ZmSAP2* and *ZmSAP4* had only one exon and no un-translation region (UTR), while others possessed a 5’-UTR and 3’-UTR, respectively (Figure 4).

The *cis*-acting elements analysis exhibited that seven kinds of *cis*-elements associated with stress response were identified in *ZmSAP* promoters (Figure 5). Among these *cis*-elements, ARE (anaerobic response element) was the most abundant *cis*-element. There were six and five AREs in the *ZmSAP3* and *ZmSAP6* gene promoters. Meanwhile, except *ZmSAP7* and *ZmSAP9*, the other eight *ZmSAP* genes contained at least one MBS (Myb binding site) element. In addition, eight kinds of hormone-responsive *cis*-elements were observed in *ZmSAP* promoters and associated with different hormones, including ABA (ABRE), ethylene (ERE), MeJA (CGTCA-Motif), salicylic acid (TCA element), auxin (TGA element) and gibberellin (P-box, GARE-motif and TATC-box) response elements. This suggests that the *ZmSAP* genes may play different roles in stress and hormone response.

### 2.5. Expression Patterns of ZmSAPs

The expression profile of *ZmSAPs* in different development stages of maize was analyzed using RNA-seq data. We found that *ZmSAPs* showed no tissue specificity in the transcription level in maize, and *ZmSAP6* exhibited a high expression level in all tissues. (Appendix A). To investigate the response of *ZmSAP*s to external stimuli, the expression profiles of 10 *ZmSAP* genes under osmotic stresses, including drought and salt, and hormone induction (ABA) were studied by RT-qPCR. Under drought stress mimicked by PEG treatment, the transcription levels of *ZmSAP2*, *ZmSAP3*, *ZmSAP5* and *ZmSAP8* were significantly up-regulated at 6, 12, 12 and 12 h of treatment, respectively. While *ZmSAP1*, *ZmSAP4*, *ZmSAP6*, *ZmSAP7* and *ZmSAP9* were significantly down-regulated by drought stress (Figure 6). In the process of salt stress, the transcription level of all *ZmSAPs* was inhibited by NaCl stress, and *ZmSAP1*, *ZmSAP2*, *ZmSAP5*, *ZmSAP7*, *ZmSAP8*, *ZmSAP9* and *ZmSAP10* showed a down-regulated during the treatment process (Figure 7). Under the induction of exogenous ABA, the expression of *ZmSAP3*, *ZmSAP4*, *ZmSAP6* and *ZmSAP8* was significantly induced by ABA at 12, 24, 24 and 24 h of treatment, respectively. However, they reached the lowest transcription level at 9 h of treatment. The transcription levels of *ZmSAP1*, *ZmSAP2*, *ZmSAP5*, *ZmSAP7*, *ZmSAP9* and *ZmSAP10* were significantly inhibited by ABA, especially *ZmSAP9* (Figure 8). These results suggest that *ZmSAPs* may play an important role in osmotic stress response.

### 2.6. Overexpression of ZmSAP2 and ZmSAP7 Enhanced the Saline Tolerance in Yeast

To validate the function of *ZmSAPs* in osmotic stresses, each *ZmSAP* gene was heterologously expressed in *Saccharomyces cerevisiae* W303a cells to phenotype on plates supplemented with mannitol or NaCl. The results showed no significant difference between yeast cells carrying pYES2-*ZmSAPs* and pYES2 (control) plasmid on the plates with mannitol (Appendix A). As shown in Figure 9A, under the plates without NaCl for control, 0.5 and 1.0 M NaCl, the yeast strain with every *ZmSAP* gene showed no difference compared to the yeast strain with empty vector pYES2, although the growth of all yeast was slightly inhibited by 1.0 M NaCl. On the plates with 1.5 M NaCl, the growth of yeast was severely inhibited. However, the yeast strains expressing *ZmSAP2* and *ZmSAP7* showed preferential growth vigor than that of pYES2 and other *ZmSAPs*. Subsequently, the yeast cells harboring *ZmSAP2* and *ZmSAP7* were cultured in liquid YNB-Ura-Gal 2% medium supplemented with 1.5 M NaCl and used to measure the growth curves. The results showed that the yeast strains with pYES2-*ZmSAP2* and pYES2-*ZmSAP7* exhibited a higher growth speed than that of pYES2 after 12 h to 72 h. The OD_600_ of them was significantly higher than the control. (Figure 9B). These results confirmed that expression of the *ZmSAP2* and *ZmSAP7* genes provide the yeast with the ability to tolerate saline stress.

## 3. Discussion

SAPs, a kind of zinc-finger protein, have been reported to be involved in multiple stress responses in plants and the immune system in humans [5,6,7,9]. Hence, the *SAP* genes are identified through genome-wide analyses from a few monocot and dicot plants such as *Arabidopsis*, rice, soybean, tomato, cotton, apple, *Brassica napu*s, cucumber, castor bean [11,35,36,37,38,39,40,41]. However, the *SAP* genes in maize were rarely reported. In the study, 10 *ZmSAP* genes were identified in maize (Table 1). The number of *ZmSAPs* shows a great deal of variation with SAPs in other plants, such as *Brassica napus* with 57 *BnSAPs*, *Glycine max* with 27 *GmSAP*s and *Populus trichocarpa* with 19 *PtSAPs* [20,35,39]. Likewise, it’s reported that there were at least 11 *ZmSAPs* in maize [42]. This should be due to the updated genome used in the present study and gene duplications resulting in variation of *SAP* numbers [38,39,40]. We also found that one pair of paralogous *ZmSAPs* and fourteen orthologous pairs between maize and rice were identified as orthologs (Figure 3 and Appendix A).

Previous studies showed that a significant majority of the *SAP* genes found in various plants had a trait of being intron-less. For instance, most *SAP* genes in rice, soybean, tomato, cucumber and castor bean possessed no intron, and only a small number of *SAP* genes contain a few introns in their gDNA [11,35,36,40,41]. Similarly, in maize, there were nine *ZmSAPs* without intron, and only *ZmSAP9* was identified by a single intron and two exons in gDNA (Figure 4). It has been confirmed that intronless genes (no introns) and intron-poor genes (three or fewer introns per gene) were more likely to play roles in osmotic stress response, including drought and salt stress, compared with intron-rich genes [43]. The genes with fewer introns could be rapidly regulated during stress and well confer the potential to establish a more quick and accurate response to stimuli by reducing the number of steps required for post-transcriptional processing [43,44]. SAP proteins are characterized by containing A20 or AN1 domains. In the study, ZmSAP2 contained one A20 domain, ZmSAP3 and ZmSAP9 richen two AN1 domains, and other ZmSAPs possessed one A20 and AN1 domain, respectively (Figure 1). These findings suggest that *ZmSAP* genes may function in the quick response to abiotic stress.

Stress-related genes can be regulated by environmental stimuli and require *cis*-elements in promoter regions to drive their transcription [45]. Herein, the composition of *cis*-elements affects gene expression and is crucial for the transcriptional control of plant growth, development, and various stress responses [46,47]. In the *ZmSAP*s promoter, different *cis*-acting elements, such as MBS, ABRE, CGTCA-motif, TGA-element, and ARE, were found and linked to responses to abiotic stimuli and hormone response (Figure 5). The MBS and ABRE elements are involved in drought and ABA response [46]. As well known, ABA acts as a crucial phytohormone and regulates plant development and stress responses [32,48]. Available reports have demonstrated that ABA can induce or inhibit the expression of several *SAP* genes, including *AtSAP9*, *AtSAP13*, *OsSAP1* and *GmSAP16*, which influence the expression of stress-related genes to respond to stress [10,18,25,35]. Here, we found that the transcription level of *ZmSAP* genes was significantly changed under osmotic stress, including drought and salt and ABA induction (Figure 6, Figure 7 and Figure 8). These findings further imply the potential roles of *ZmSAPs* in stress response.

In yeast cells, the heterologous expression of *ZmSAPs* did not confer yeast tolerance to drought mimicking by mannitol (Appendix A). Only *ZmSAP2* and *ZmSAP7* significantly improved yeast tolerance to high salt (Figure 9). The functional differentiation of *ZmSAPs* can be explained by their evolutional diversification with uneven distribution and duplication on chromosomes and different composition of the domain (Figure 1 and Figure 2) [49]. Interestingly, the expression of *ZmSAP2* and *ZmSAP7* was inhibited by salt in maize (Figure 7). Similarly, the phenomenon is found in previous studies. The expressions of *CaSAPs* are up-regulated by low temperature and dehydration stress, but *CaSAPs*-silenced pepper plants show tolerance to low temperature and drought [50]. Likewise, the maize *ZmBES1/BZR1-5* is inhibited by drought, but its overexpression improves drought tolerance in transgenic *Arabidopsis* [51]. It can be explained by its unknown upstream regulators and elucidated in further study. 

Moreover, it has been proven that heterologous expression of the *SAP* gene from *Aeluropus littoralis* (*AlSAP*), *Lobularia maritima* (*LmSAP*), *Leymus chinensis* (*LcSAP*) also enhanced cell tolerance to salt, ionic and osmotic stresses in yeast [21,52,53]. More importantly, *SAPs* have exhibited functional diversity in osmotic stress, including drought and salt. For instance, *OSISAP1*, *OsiSAP1*, *OsiSAP8* and *AtSAP5*, as positive regulators, improved drought, salt, and osmotic tolerance in transgenic tobacco, rice and *Arabidopsis*, respectively [10,16,17,54]. However, some *SAP* genes play a negative role in stress tolerance. For example, rice *OsiSAP7* and *ZFP185* negatively regulate ABA stress signaling and impart sensitivity to drought and salt stress in transgenic *Arabidopsis* and rice [30,33]. Likewise, the downregulation of *PagSAP1* in poplar significantly enhances tolerance to salt stress, increases the K^+^/Na^+^ ratio in roots, and alters gene expression related to cellular ion homeostasis [55].

In conclusion, a total of 10 *ZmSAP* genes were identified in the maize genome. All ZmSAPs belong to the family of zinc-finger proteins with the A20/AN1 domain. The expression of *ZmSAP* genes was regulated by osmotic stresses, including drought and salt, as well as ABA. Furthermore, heterologous expression of *ZmSAP2* and *ZmSAP7* significantly improved the saline tolerance in yeast cells. The study suggests that *ZmSAPs* may play important roles in response to abiotic stresses and provides insights into further underlying the regulatory mechanisms of *ZmSAPs* in regulating stress response in maize.

## 4. Materials and Methods

### 4.1. Identification of ZmSAPs in Maize

The genome and protein sequence data (reference 5.0) of maize B73 were retrieved from the MaizeGDB database (https://download.maizegdb.org/Zm-B73-REFERENCE-NAM-5.0/, accessed on 15 October 2021) and used for ZmSAPs search. Subsequently, the 14 and 18 SAP protein sequences of *Arabidopsis* and rice were downloaded from the *Arabidopsis* Information Resource (http://www.arabidopsis.org/, accessed on 15 October 2021) and the Rice Genome Annotation Project (http://rice.plantbiology.msu.edu/index.shtml, accessed on 15 October 2021) and used as queries to perform local BLASTp search against above maize database by using the BLAST + suite with an E-value 1e^−10^, respectively [11]. Meanwhile, the A20 domain (Pfam ID: PF01754) and AN1 domain (Pfam ID: PF01428) were downloaded from the Pfam database (https://pfam-legacy.xfam.org/, accessed on 15 October 2021) and used to further identify ZmSAP candidates using HMMER3.0. Then, the redundant sequences were removed manually to acquire ZmSAPs.

According to the method described by Yu et al. [49], the amino acid sequences of ZmSAPs were used to analyze physical and chemical properties and secondary structure by using ProtParam (http://web.expasy.org/protparam/, accessed on 20 October 2021), GOR IV (http://npsa-pbil.ibcp.fr/cgi-bin/npsa_automat.pl?page=npsa_gor4.html, accessed on 20 October 2021), TMHMM Server v. 2.0 (https://services.healthtech.dtu.dk/service.php?TMHMM-2.0/, accessed on 20 October 2021) and SignalP 4.1 (https://services.healthtech.dtu.dk/service.php?SignalP-5.0/, accessed on 20 October 2021), respectively. The subcellular location was predicted by using WoLF PSORT (http://www.genscript.com/wolf-psort.html, accessed on 20 October 2021). The conserved domains and motifs of ZmSAPs were further analyzed via searching the Conserved Domain Search Database (CDD, http://www.ncbi.nlm.nih.gov/Structure/cdd/cdd.shtml, accessed on 20 October 2021) and using Multiple Em for Motif Elicitation (MEME) online program (https://meme-suite.org/meme/doc/meme.html, accessed on 20 October 2021) [56], respectively.

The amino acid sequence of AtSAPs and OsSAPs were used for phylogenetic analysis against with ZmSAPs of maize. The phylogenetic tree was constructed using MEGA7.0 with the neighbor-joining (NJ) method (http://www.megasoftware.net, accessed on 28 October 2021) with 1000 bootstrap replications. The phylogenetic tree among ZmSAP members was also constructed using the same method.

### 4.2. Chromosomal Location, Gene Replication and Structure analysis

The physical location of *ZmSAP*s was obtained from position information in the MaizeGDB database. Subsequently, *ZmSAP*s were mapped to maize chromosomes by using Circos [37]. The gene replication events among *ZmSAP* genes were analyzed using the Multiple collinear scanning toolkits (MCScanX) with the default parameters [57]. The synteny relationship among *SAP* genes of maize, *Arabidopsis* and rice was analyzed using Dual Synteny Plotter software (https://github.com/CJ-Chen/TBtools, accessed on 10 November 2021) [58]. The cDNA and genomic sequences of *ZmSAP*s were obtained from MaizeGDB and then used to analyze the exon–intron organizations and intron type by using Gene Structure Display Server (GSDS) (http://gsds.gao-lab.org/, accessed on 10 November 2021).

The 2000 bp upstream of translation start site (TSS) of *ZmSAPs* were downloaded from MaizeGDB and used for *cis*-elements analysis by using PlantCARE online software (available online: http://bioinformatics.psb.ugent.be/webtools/plantcare/html/, accessed on 15 November 2021).

### 4.3. Tissue Expression Analysis of ZmSAPs

To examine the potential role of *ZmSAPs* in maize development, the expression feature of *ZmSAPs* in different development stages was analyzed. The RNA-seq data of *ZmSAPs* in different tissues were obtained from MaizeGDB qTeller center (https://qteller.maizegdb.org/, accessed on 5 November 2022), displayed as fragments per kilobase of the exon model per million mapped fragments (FPKM) values and visualized as a heatmap using TBtools [56].

### 4.4. Plant Materials and Stress Treatments

The seeds of the maize inbred line B73 were surface-sterilized and germinated in filter paper for 48 h. The seedlings were transplanted into a Hoagland’s solution for a hydroponic culture under 16 h light at 28 °C/8 h dark at 25 °C periods. At the three-leaf stage, the seedlings with the same size were divided into four groups. Each of the three groups of seedlings was treated with 16% PEG-6000, 250 mM NaCl and 100 μM ABA solution, respectively. One group of seedlings was used as a control without treatment. At 0, 3, 6, 9, 12, and 24 h of treatment, the leaves were sampled, frozen and ground in liquid nitrogen, and stored at −80 °C for an RNA extraction, with three replicates.

### 4.5. QRT-PCR Analysis

The total RNA of every sample was extracted by using RNAiso plus kit (TaKaRa, Dalian, China) according to the manufacturer’s instruction, then treated with RNase-free DNase, and quantified using NanoDrop^TM^ One^C^ (ThermoScientific, Waltham, MA, USA). Subsequently, the 100 ng RNA of every sample was reverse transcribed into cDNA using the PrimeScript^TM^ reagent kit (TaKaRa, Dalian, China). The cDNA samples were stored at −20 °C and used for quantitative real-time PCR (qRT-PCR).

The specific primer pairs of *ZmSAPs* and *ZmEF-1a* for internal control were designed by using the Primer-BLAST (http://www.ncbi.nlm.nih.gov/tools/primer-blast, accessed on 5 January 2022) and synthesized at TSINGKE (China) (Appendix A). As described by Sun et al. [52], the qRT-PCR was conducted in CFX96^TM^ Real-Time System (BioRad, Hercules, CA, USA) using SYBR PremixEx Taq^TM^ II (TaKaRa, Dalian, China) with three technical replicates. The expression of *ZmSAP* genes was analyzed and normalized by using the 2^−ΔΔCT^ method of the CFX Manger^TM^ software version 2.0 (Bio-Rad, Hercules, CA, USA) [59]. The data were shown as mean value ± standard deviation (SD). The statistical significance among three biological replicates was tested by the Student’s *t*-test.

### 4.6. Stress Tolerance Test of ZmSAPs in Yeast Cells

The specific primer pairs of ZmSAPs were designed using Primer 5, synthesized at TSINGKE (China) (Appendix A), and used to amplify the open reading frame (ORF) of *ZmSAPs* from cDNA of maize B73 inbred line by PCR amplification. The purified PCR product of every *ZmSAP* was inserted into the *BamH*I/*Xho*I site of the pYES2 vector (INVITROGEN, Waltham, MA, USA). The recombinant plasmid and empty vector pYES2 were transformed into *Saccharomyces cerevisiae* W303a (*MATa ade2 ura3 leu2 his3 trp1*) by standard PEG lithium acetate method, respectively [60]. The transformed yeast solution was spread on the yeast nitrogen plates lacking uracil (YNB-Ura) and cultured at 28 °C in an incubator for 2–3 days. The positive colony transformed by every *ZmSAP* was identified by PCR and incubated overnight in liquid YNB-Ura medium to OD_600_ to 1.0. 

According to the methods of Ben et al. [52,53], with minor modification, the yeast cultures were diluted to successive gradient dilutions (10^−1^, 10^−2^, 10^−3^, 10^−4^) using liquid YNB-Ura medium. Then, 8 μL of each diluent was placed on YNB-Ura-Gal 2% solid medium supplemented with 0.5, 1.0, 1.5 M NaCl or 2.0, 2.5, 3.0 M Mannitol, respectively, and cultured at 28 °C in an incubator for 2–3 days for phenotyping. Subsequently, the candidates showing preferential growth vigor on solid medium were selected for dissecting the growth curve. The OD_600_ of yeast line with candidate genes was adjusted to 0.2, then 1 mL of them was added into 20 mL YNB-Ura-Gal 2% liquid medium, and cultured at 28 °C. At 0, 12, 24, 36, 48 and 72 h, the OD_600_ of every culture was monitored with three replicates. The yeast transformed by the pYES2 vector was used as a control. The 2% galactose (Gal) was added into the YNB-Ura medium to induce the expression of *ZmSAPs* under the control of the *Gal* promoter.

## Figures and Tables

**Figure 1 ijms-23-14010-f001:**
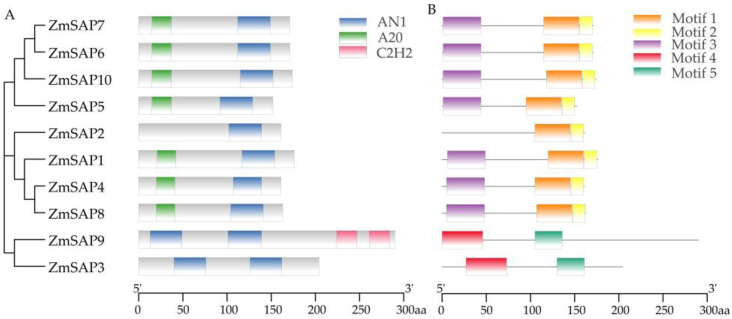
The schematic diagram of the conserved domain and motif composition of ZmSAP members. (**A**) The domain composition. (**B**) The motif composition.

**Figure 2 ijms-23-14010-f002:**
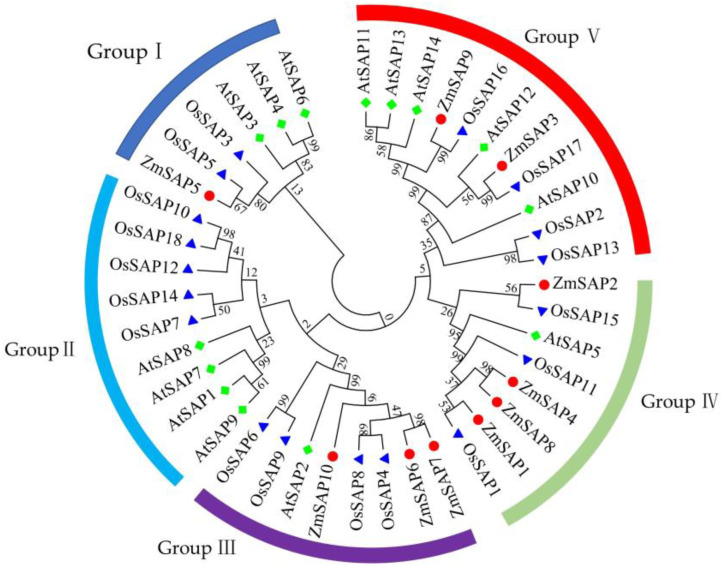
Phylogenetic evolutionary tree of SAP family in maize, *Arabidopsis* and rice. The red dot indicates the ZmSAP, the blue triangle indicates the OsSAPs, and the green diamond represents the AtSAPs.

**Figure 3 ijms-23-14010-f003:**
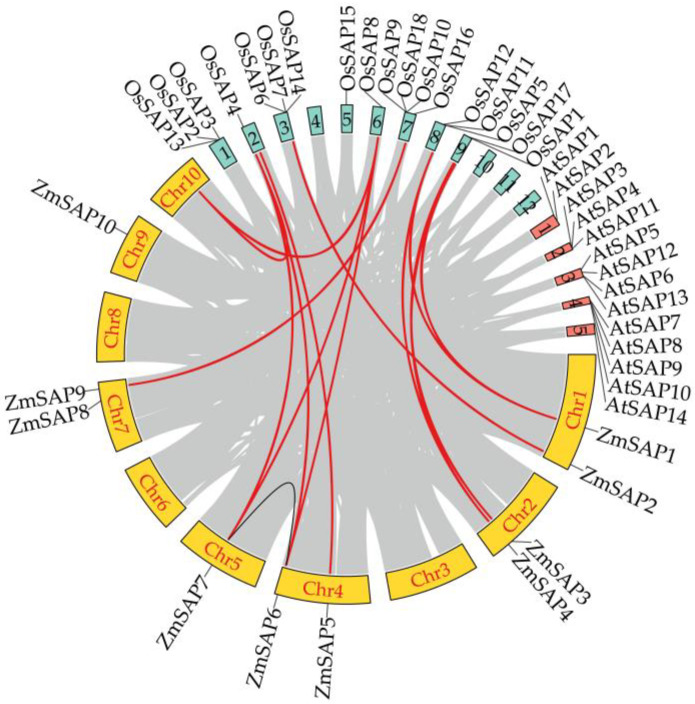
Chromosome localization and collinearity analysis of *SAP*s in maize. The inner black line represents the paralogous pair of *ZmSAPs*. The inner red line indicates collinearity within species of *SAP* genes between maize and rice. The yellow blocks represent 10 chromosomes of maize, the green blocks represent 12 chromosomes of rice, and the red blocks represent 5 chromosomes of *Arabidopsis*.

**Figure 4 ijms-23-14010-f004:**
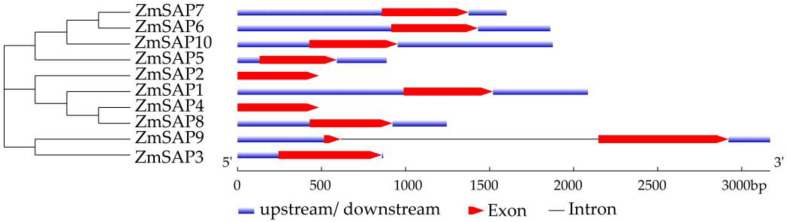
Gene structure of *ZmSAPs*. The red triangles indicate exons, and the blue boxes indicate 5’ or 3’ UTR, and the black line connecting exons indicates an intron.

**Figure 5 ijms-23-14010-f005:**
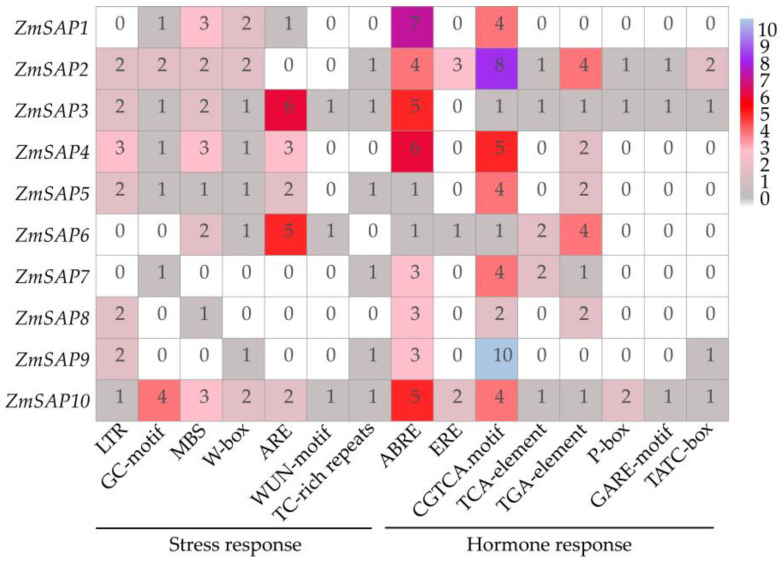
The *cis*-acting elements *ZmSAP* promoter region. The numbers in the blocks represent the number of *cis*-acting elements. LTR—low-temperature response; GC-motif—enhancer-like element involved in anoxic specific inducibility; MBS—(Myb binding site) drought response; ARE—(anaerobic response element) anaerobic induction; TC-rich repeats and W-box—defense and stress response; WUN-motif—wounding response; ABRE (ABA response element)—ABA response; ERE (ethylene response element)—ethylene response; CGTCA-Motif—MeJA response; TC-element—salicylic acid response; TGA-element—auxin response; P-box—GARE-motif and TATC-box—gibberellin response.

**Figure 6 ijms-23-14010-f006:**
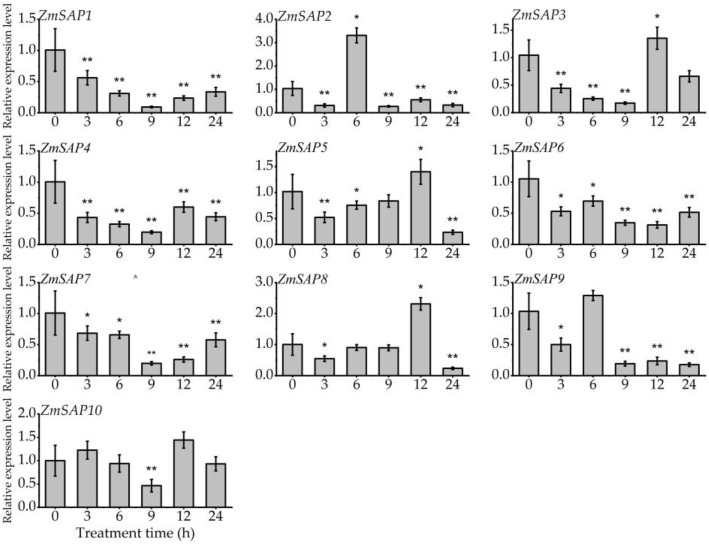
The expression of *ZmSAP*s under drought stress mimicked by 16% PEG-6000 treatment. * and **, indicates significant differences at *p* < 0.05 and *p* < 0.01, respectively.

**Figure 7 ijms-23-14010-f007:**
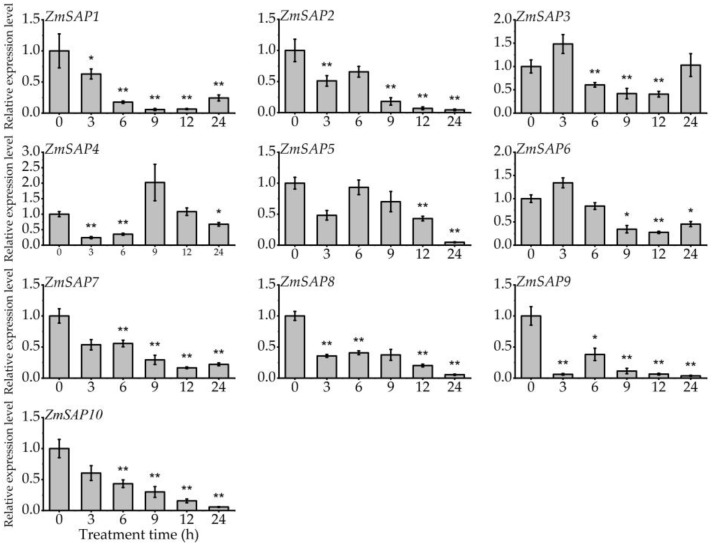
The expression of *ZmSAP*s under salt stress by 250 mM NaCl treatment. * and **, indicates significant differences at *p* < 0.05 and *p* < 0.01, respectively.

**Figure 8 ijms-23-14010-f008:**
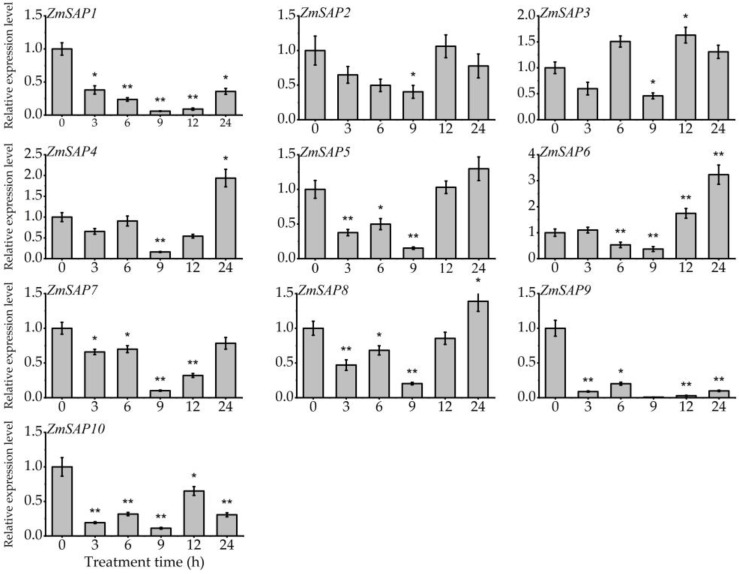
The expression of 10 *ZmSAP* genes in response to 100 μM ABA treatment. * and **, indicates significant differences at *p* < 0.05 and *p* < 0.01, respectively.

**Figure 9 ijms-23-14010-f009:**
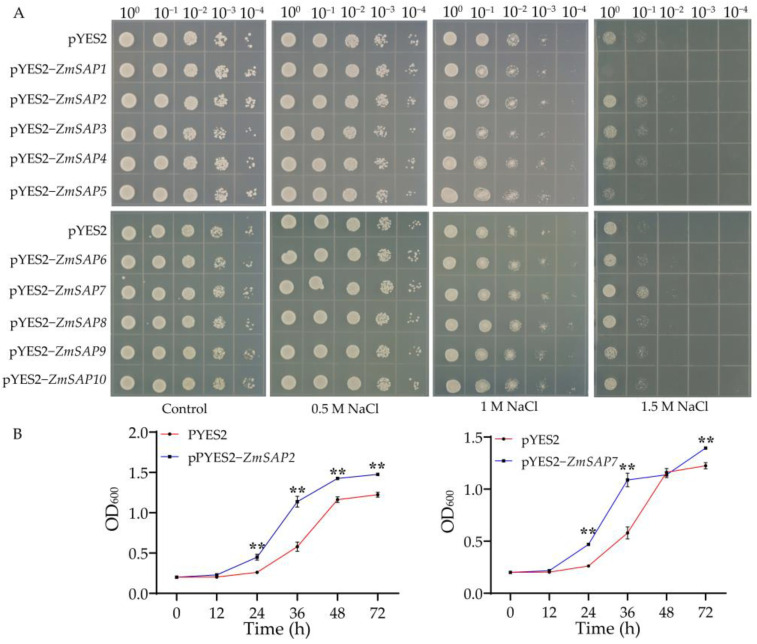
The phenotype of yeast cells carrying *ZmSAP* genes under salt stress by NaCl treatment. (**A**) The phenotype of all yeast strains on solid YNB-Ura-Gal 2% medium without (control) or with 0.5, 1.0 and 1.5 M NaCl. Photographs were taken after four days of incubation at 28 °C. (**B**) The growth curve of yeast cells expressing pYES2, pYES2-*ZmSAP2* and pYES2-*ZmSAP7* plasmid in YNB-Ura-Gal 2% liquid medium supplemented with 1.5 M NaCl for three days at 28 °C with an initial OD_600_ = 0.2, respectively. **, indicates significant differences at *p* < 0.01.

**Table 1 ijms-23-14010-t001:** Characteristics of *ZmSAP* genes in *Zea mays*.

Gene ID	Gene Name	Number of Amino Acids	Molecular Weight (KDa)	pI	CDS (bp)	GC (%)	Grand Average Hydropathy	Subcellular Locations	Instability Index
Zm00001eb034760	*ZmSAP1*	176	18.75	9.12	531	73.07	−0.572	N	68.93
Zm00001eb060270	*ZmSAP2*	161	16.68	9.53	486	65.02	−0.515	N	54.33
Zm00001eb099020	*ZmSAP3*	204	22.15	9.39	615	71.22	−0.713	N	48.66
Zm00001eb101840	*ZmSAP4*	161	16.78	9.19	486	73.25	−0.276	N	50.93
Zm00001eb181400	*ZmSAP5*	152	16.00	9.01	459	70.81	−0.387	N	51.56
Zm00001eb205990	*ZmSAP6*	171	18.31	8.28	516	54.84	−0.323	N	31.15
Zm00001eb236360	*ZmSAP7*	171	18.29	8.28	516	56.59	−0.235	N	30.92
Zm00001eb316600	*ZmSAP8*	163	17.20	9.45	492	73.37	−0.458	N	62.8
Zm00001eb324750	*ZmSAP9*	290	32.04	8.58	873	49.37	−0.59	N	38.37
Zm00001eb388350	*ZmSAP10*	174	18.41	8.48	525	59.62	−0.198	N	26.45

Note: N stands for nucleus.

## Data Availability

Not applicable.

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
