# Peer review of "Comprehensive Identification and Functional Analysis of Stress-Associated Protein (SAP) Genes in Osmotic Stress in Maize"

_ijms, 2022, doi:10.3390/ijms232214010_

Round 1

Reviewer 1 Report

Osmotic stress such as salt, drought seriously affect maize growth and yield. Identification of stress tolerance genes is of great significance for maize to solve these problems. The manuscript by Fu et al. identified the composition of SAP genes in maize and reported their properties including gene structure and duplication, chromosomal locations, cis-acting regulatory elements. Additionally, they found the transcription patterns of ZmSAPs was largely affected by osmotic stress and ectopic expression of ZmSAP2 and ZmSAP7 could improve the saline tolerance of yeast cells. The finding is interesting and the manuscript is well organized and written. However, there are still some minor issues need to be addressed in the manuscript before it is recommended to publish.

1. To examine the potential role of ZmSAPs in maize development, the expression feature of ZmSAPs in different development stages should be analyzed. The authors mentioned tissue expression profiles in Line 74.

2. In results part, Line 132-134, what did “... SAPs might occur high frequency of duplication between rice and maize” mean? It is ambiguous and I suggest the author to rewrite this part.

3. There were multiple cis-acting elements identified in ZmSAP promotors. Was there any correlation between the expression patterns and the composition of cis-acting elements of ZmSAPs?

4. ZmSAP2 and ZmSAP7 were obviously repressed under osmotic stress in maize, however, yeast cells carrying ZmSAP2 and ZmSAP7 genes showed enhanced salt stress tolerance. I suggest the author to discuss their relationship in Discussion part.

Reviewer 2 Report

Stress-associated proteins (SAPs) are zinc finger proteins involved in the regulation of various stresses in a variety of plant species. Using the maize genome, a total of 10 ZmSAP genes were identified by the authors. The A20/AN1 domain-containing family of zinc-finger proteins comprises all ZmSAPs. Osmotic stressors, including as dryness and salt, as well as ABA, controlled the expression of the ZmSAP genes. Additionally, ZmSAP2 and ZmSAP7 heterologous expression dramatically increased yeast cell saline tolerance. The study offers more insights into the regulatory mechanisms of ZmSAPs in controlling the stress response in maize and implies that ZmSAPs may play significant roles in the response to abiotic stresses. However, there are several shortcomings in the findings that prevent the manuscript from making an impactful contribution.

1.     My only major concern is regarding stress treatments. The author mentioned that they used different concentrations and time points to evaluate the effect of abiotic stress. But I couldn’t find such results in the figure. No concentrations are mentioned, only one result is shown

2.     Although English is readable and understandable, it is still mind-taxing. The sentence order is in bad form. Language proofing is definitely required. The punctuation marks, full stops, and other minor grammatical errors must be thoroughly checked. Some sentence requires rephrasing. For example, lines, 16-18; 26-27

3.     Fig.2, which phylogenetic tree? How the tree was constructed, should be mentioned

Author Response

Response to reviewers

Dear Editors and Reviewers:

Thank you very much for your comments concerning our manuscript. Those comments are very valuable and helpful for revising and improving our paper, as well as provide important guiding significance to our study. We have studied all comments carefully and have addressed every point. The responds to comments are as follows:

Reviewer 2

Stress-associated proteins (SAPs) are zinc finger proteins involved in the regulation of various stresses in a variety of plant species. Using the maize genome, a total of 10 ZmSAP genes were identified by the authors. The A20/AN1 domain-containing family of zinc-finger proteins comprises all ZmSAPs. Osmotic stressors, including as dryness and salt, as well as ABA, controlled the expression of the ZmSAP genes. Additionally, ZmSAP2 and ZmSAP7 heterologous expression dramatically increased yeast cell saline tolerance. The study offers more insights into the regulatory mechanisms of ZmSAPs in controlling the stress response in maize and implies that ZmSAPs may play significant roles in the response to abiotic stresses. However, there are several shortcomings in the findings that prevent the manuscript from making an impactful contribution.

  1. My only major concern is regarding stress treatments. The author mentioned that they used different concentrations and time points to evaluate the effect of abiotic stress. But I couldn’t find such results in the figure. No concentrations are mentioned, only one result is shown

Answer: Thanks for your comments to improve our manuscript. We revised the figure legend of figures 6, 7 and 8. The results of yeast phenotype under drought stress were shown in Figure S4 in the supplementary files. We’re sorry for forgetting to upload the attached file in the previous version.

  1. Although English is readable and understandable, it is still mind-taxing. The sentence order is in bad form. Language proofing is definitely required. The punctuation marks, full stops, and other minor grammatical errors must be thoroughly checked. Some sentence requires rephrasing. For example, lines, 16-18; 26-27

Answer: Thanks for your careful comments. We asked a native English speaker to revise the language.

  1. 2, which phylogenetic tree? How the tree was constructed, should be mentioned

Answer: It’s a phylogenetic evolutionary tree of SAP family in maize, Arabidopsis and rice. The method was described in the last paragraph of Materials and Methods 4.1 (Line326-330).

Round 2

Reviewer 2 Report

The authors have resolved all my queries. The manuscript can be accepted now